# Pre-Layout Parasitic-Aware Design Optimizing for RF Circuits Using Graph Neural Network

**Chenfeng Li [1,2,\*], Dezhong Hu [1,2] and Xiaoyan Zhang [1,2]**

1   School of Electronic Science & Engineering, Southeast University, Nanjing 210096, China
2   National ASIC Center, Southeast University, Nanjing 210096, China
\*   Correspondence: 230169086@seu.edu.cn

**Abstract:** The performance of analog and RF circuits is widely affected by the interconnection parasitic in the circuit. With the progress of technology, interconnection parasitics plays a larger role in performance deterioration. To solve this problem, designers must repeat layout design and validation process. In order to achieve an upgrade in the design efficiency, in this paper, a Graph Neural Network (GNN)-based pre-layout parasitic parameter prediction method is proposed and applied to the design optimization of a 28 nm PLL. With the new method adopted, the frequency band overlap rate of the VCO is improved by 2.3 percents for an equal design effort. Similarly, the optimized CP is superior to the traditional method with a 15 ps mismatch time. These improvements are achieved under the premise of greatly saving the optimization iteration and verification costs.

**Keywords:** analog circuit; machine learning; graph neural network; RF circuit

## 1. Introduction

In the design and optimization of integrated circuits, parasitic effects between wires, devices, and the environment must be considered. These parasitic effects are not specifically described in the device model because parasitic effects depend heavily on the floorplan in the layout. This is especially the case in an RF circuit design, in which high-frequency signals are threatened by the surrounding signal wires and large-scale spiral inductance nearby. With the size of the device shrinking to the nanometer scale, the parasitic effect on the circuit becomes even more significant.

The pre-simulation does not include interconnect parasitism, which results in a huge difference between the performance obtained from the pre-simulation and post-simulation. This causes many iterations between the pre-layout and post-layout phases, which leads to the realignment of the circuit. For complex designs, the process could be time-consuming, typically taking days to weeks. If the performance after layout and wiring can be predicted with a certain accuracy in the circuit design stage, the number of iterations in the process from schematic to layout and back to schematic can be greatly reduced, so as to improve the efficiency of circuit design.

In order to solve the problem of considering parasitic information in the pre-layout stage, Hiroaki Yoshida et al. [1] predicted layout parasitism and device parameters in submicron technology. However, the accuracy of this approach depends heavily on the estimation of the maximum transistor sequence (MTS), which is a term relating to whether a transistor is a shared source or drain diffusion. Brett Shook [2] et al. proposed a parasitic estimation framework based on machine learning, called MLParest. MLParest provides an accurate estimate of the interconnect parasitic resistance and capacitance in the pre-layout design phase. However, this method inputs features that lack the global information of the circuit structure, which leads to an unsatisfactory prediction. ParaGraph [3] is a learning-based framework that is designed to predict layout-dependent effects and device parameters, including parasitics. The output of the prediction, however, is not adopted

into the design improvements of the circuit, and the heterogeneous embedding method is not suitable for the RF design.

In this paper, we present a predictive-based approach to accelerate the iterative design of analog circuits. By constructing a GNN network [4] with supervised learning, and inputting netlist information before the physical design, the parasitic capacitance and resistance of the key nodes in the circuit can be quantitatively predicted. Thus, this method can greatly reduce the time cost in design optimization. To validate this optimization framework, we demonstrate the efficiency of the process by optimizing a 2.4 GHz charge-pump PLL circuit.

The content of this paper is as follows: In Section 2, the main circuit parasitic prediction framework is introduced, along with how to realize parasitic parameter prediction before the layout is obtained. In Section 3, a PLL circuit design is presented as the background case, two circuits that can explain the advantages of this process are analyzed, and we explain why and how the parasitic parameter prediction is applied. The 4th section describes the specific optimization method, and presents some results that compare the efficiency and final performance with the traditional iterative method of front and back-end design. Finally, we give a summary, and evaluate the performance of the prediction model itself and the performance gained in the whole optimized flow.

## 2. GNN-Based Model for Interconnect RC Prediction

The structure of the prediction model is shown in Figure 1. The method of data processing, training, and prediction process will be discussed in detail. Given the input netlist, each net's parasitic resistance and capacitance parameters are predicted and attached to the corresponding entity in the structure as an output.

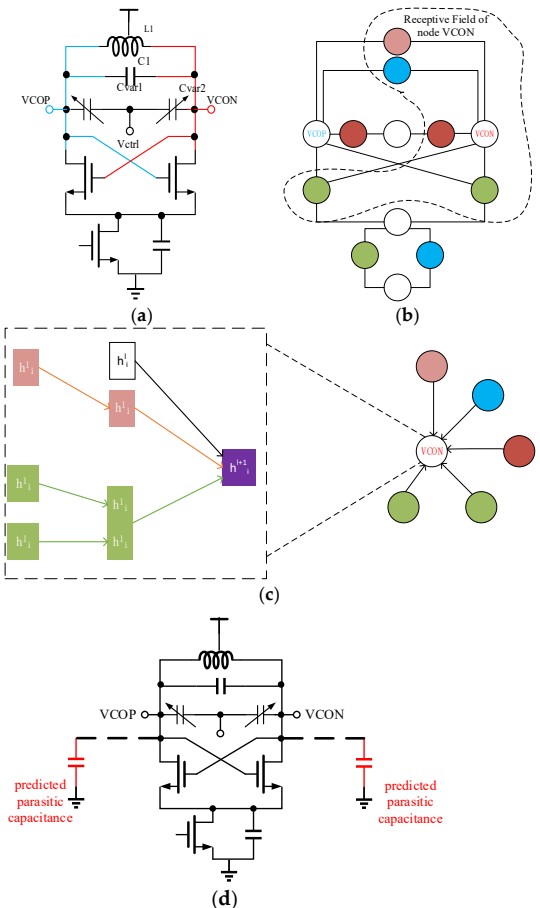

**Figure 1.** GNN-Based parasitic prediction. (**a**) Pre-layout schematics; (**b**) Graph embedding; (**c**) Message passing and Node aggregation; (**d**) Schematics with predicted parasitics.

Without physical design and layout information, the proposed model can predict certain layout and parasitic parameters. The theoretical basis of this prediction model is to predict the behavior in the floorplan. Due to space limitations, the topological relationships in the netlist are usually related to the physical floorplan, i.e., floorplan and routing. Thus, the parasitic information should be able to be predicted given the information before the layout is obtained, that is, the parameters of each device in the netlist and how they are connected.

### 2.1. Inputs and Outputs of the Model

To make the tool compatible with the data form in the present EDA design flow, the netlist is selected as the input data form for graph embedding. After preprocessing, an embedded graph is taken as the input of the model for training, and the parasitic information extracted from the layout is also taken as the input for training data, as shown in Table 1.

**Table 1.** Embedded features from the netlist.

| Node Type | Features | Definition |
| --- | --- | --- |
| PMOS/NMOS * | W | width |
| | L | length |
| | M | multiplier |
| | N | Number of fingers |
| Capacitor * | W | width |
| | L | length |
| Resistor * | W | width |
| | L | length |
| | M | multiplier |
| Nets ** | PC | Predicted capacitor |
| | PR | Predicted resistance |

* capacitors and resistors have different realizations in each tech library, and they are modeled into different categories of nodes in case the materials or basic structure of the device are different. e.g., cfmom and nmoscap.
** fanout is not considered an attribute of the nets as they are exploited in the topology of the graph.

A variety of RC topologies can be obtained by extracting parasitic parameters from a single-wire network. In contrast, the traditional RC extraction tool can obtain a rather complex RC network. For our work, the same complexity is neither necessary nor feasible. In previous work [2], only a simple RC structure is given for each network in the prediction results of parasitic information. This choice has several advantages: it simplifies the calculation and modeling complexity, and the simplicity of the RC network can help estimate the circuit performance by simulating with enough accuracy.

There are two major strategies to map a netlist to a graph, i.e., embedding. One choice is to model circuit devices as nodes and nets as edges. Another is to model both the devices and the interconnects as different nodes. In this work, considering that both nodes or nets contain essential information, they are all modeled as nodes. Therefore, the edges of the graph in this work include the connection of various networks, depending on the type of node it connects, e.g., mosfet_drain_to_net, net_to_mosfet_drain, etc.

After the embedding, the graph represents the circuit structure giving the netlist. The device parameters are represented by specific data attached to the nodes, as shown in the following table.

### 2.2. Graph Neural Network

The network is composed of multiple GNN layers [4], which output with the same topology as the input heterogeneous graph. The predicted interconnection parasitic information is generated and attached in nodes with the type of "net" in the graph.

In the connection layer, the input graph is processed by an MLP (multilayer Perceptron) to generate the output. Inside the MLP, the information in the nodes is passed along the edges, and the received message is then combined with the original local tags to update itself. This process is referred to as message passing and neighbor aggregation, respectively.

In terms of message passing, traditional methods include Graph Convolutional Networks (GCN) [5], GraphSAGE [6], Relational GCN (RGCN) [7], and Graph Attention Networks (GAT) [8]. Despite that fact that they are all designed for graph-like data, only RCGN [7] is designed for heterogeneous graphs; none of them are compatible with parasitic prediction in RF designs, where the device type matters tremendously to the area it affects as it comes to the coupling capacitance.

We use the combination of two node feature vectors to represent the edge vectors' connection features, then combine the aggregated information with the central node information, which represents the embedding of a node; this is the combination method of the central node and the neighborhood node. Popular aggregation methods and the aggregation method we proposed is listed in Table 2.

**Table 2.** Aggregation methods of the present GNN models and the proposed model.

| GNN Models | Aggregation Method |
|:---:|:---:|
| GCN [5] | $h_i^{(l+1)} = \sigma\left( b^{(l)} + \sum_{j \in N(i)} \frac{1}{c_{ij}} W^{(l)} h_j^{(l)} \right)$ |
| GraphSage [6] | $h_i^{(l+1)} = \sigma\left( W \cdot \text{concat}\left( h_i^{(l)}, h_{N(i)}^{(l+1)} + b^{(l)} \right) \right)$ |
| RGCN [7] | $h_i^{(l+1)} = \sigma\left( h_{N(i)}^{(l+1)} + W_0^{(l)} h_i^{(l)} \right)$ |
| GAT [8] | $\begin{aligned} e_{ij}^l &= \vec{a}^T \text{concat}\left( W^{(l)} h_i^l, W^{(l)} h_j^l \right) \\ \alpha_{ij}^l &= \text{softmax}x_i\left( \text{LeakyReLU}\left( e_{ij}^l \right) \right) \\ h_i^{(l+1)} &= \sigma\left( \sum_{j \in N(i)} \alpha_{i,j} W^{(l)} h_j^{(l)} \right) \end{aligned}$ |
| ParaGraph [3] | $\begin{aligned} h_i^t &= \sum_{j \in N_t(i)} \alpha_{i,j} W_t^{(l)} h_j^{(l)}, \forall i \in N \\ h_i &= \sum_{t \in E_T} h_i^t, \forall i \in N \\ h_i^{(l+1)} &= \sigma\left( W^{(l)} \cdot \text{concat}\left( h_i^{(l)}, h_i + b^{(l)} \right) \right) \end{aligned}$ |
| Proposed method | $\begin{aligned} h_i^t &= \sum_{j \in N_t(i)} \alpha_{i,j} k_{i,j} W_t^{(l)} h_j^{(l)}, \forall i \in N \\ h_i &= \sum_{t \in E_T} h_i^t, \forall i \in N \\ h_i^{(l+1)} &= \sigma\left( W^{(l)} \cdot \text{concat}\left( h_i^{(l)}, h_i + b^{(l)} \right) \right) \end{aligned}$ |

Each node in the graph performs the calculation described above. Thus, each node can obtain information describing its neighbors and the global structure. Through several rounds of aggregation and embedding, each node can integrate information far from itself. After each round of calculation, the obtained features need to be processed to obtain the output as the predicted value.

GraphSAGE [6] is adopted to realize node embedding and aggregation in this model, and it is improved with the RGCN algorithm [7] and the GAT algorithm [8]; this is due to the heterogeneous nature of the RF analog circuit, which contains many different devices. In the process of aggregation, different edges are grouped independently to deal with heterogeneous information. As shown in Figure 1c, the neighborhood nodes of a specific node are mainly divided into different types, which are aggregated into node features according to different aggregation methods and the weights of the edges; finally, they are combined with the node itself to update the embedding.

The universal iteration method of GNN provides a series of information transfer paradigms for heterogeneous graphs. It is worth noting that no one has yet proposed information transfer rules for GNN models specifically for the RC prediction of Parasitic Parameters. Therefore, this work proposes a set of rules to fulfill the needs of this evaluation scenario. Compared to ParaGraph [3], while the features are being aggregated from neighbor nodes, a compensatory factor $k_{i,j}$ is joined to describe the long-range coupling effect, $k_{i,j}$ is independent from the edge-dependent attribute $\alpha_{i,j}$, and can be turned off during the graph generation stage, depending on whether the device or net is close enough to an unprotected (e.g., surrounded by a guard ring) coupling source (e.g., a spiral inductor).

### 2.3. Training and Network Optimization

In this study, the graph neural network will be used to establish and fit the relationship between the front–end design parameters and the parasitic prediction results. The process of establishment, initialization, training, and prediction basically follows the same principles as in a BP network. Basic settings of the network is listed in Table 3.

**Table 3.** Settings of the GNN training.

| Settings | Values |
| --- | --- |
| Learning rate | 0.003 |
| Hidden feats | 7 |
| Hidden num | 5 |
| Aggregator type | mean |
| Activation function | Relu |
| Feature drop num | 0 |
| Epochs | 300 |

The process of continuously adjusting the weights to minimize the error function is referred to as training. In this paper, the Adams method, namely, adaptive moment estimation, is used as the basic algorithm for parameter updating. The basic idea of the algorithm design is to maintain the stability of the gradient direction and step size in the first and second moments of the parameter iteration process, in order to improve the convergence efficiency of the algorithm and avoid oscillation and local convergence.

### 3. Experiments

The design case presented below is from an analog phase-locked loop in an RF circuit. Before the following optimization, the designer adopted a more conservative strategy for the design parameters of each module, in order to avoid the challenge of parasitic parameters. This also comes with several PPA concessions. Based on the proposed parameter design and optimization framework, we re-designed the circuit parameters in multiple modules. Specific methods and results are given below.

### 3.1. Low-Voltage Charge Pump

The low-voltage charge pump structure discussed in this paper is shown in Figure 2. Operational amplifiers A1 and A2 are used to clamp the source and drain voltages of charge and discharge transistor P1 and N1, respectively; this is performed to reduce the influence of the channel length modulation effect and maintain a high matching degree in the current replication process [9,10].

In this circuit, the interconnection parasites affect the circuit performance mainly through charging time mismatch and overshoot current. When the parasitic capacitance of net A or B increases, the arrival of charge and discharge signals will be delayed; this will slow down the charge and discharge process, potentially causing the current at the output to overshoot. The influence of capacitance at nodes A and B on the mismatch current is shown in Figure 3.

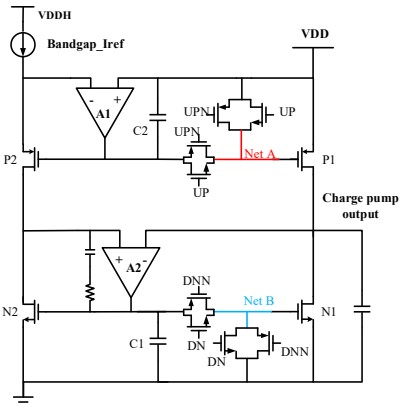

**Figure 2.** Low-voltage charge pump.

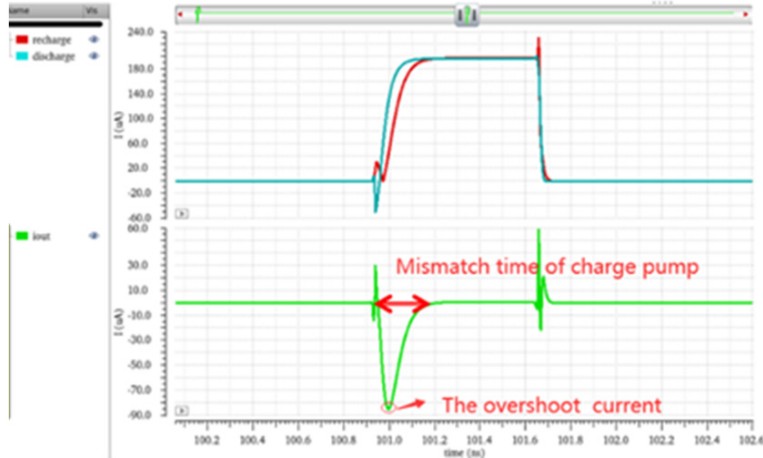

**Figure 3.** Parasitics' impacts on charge/discharge.

In a traditional design, a set of parameters is usually initially set to give the switch transistors a suitable on-resistance. For the two current-mirror transistors, to reduce their gate capacitance, we usually choose a smaller channel length, then make the charge and discharge current speed similar through gradual adjustments of the size of the transistors; this is in order to obtain a shorter adaptation time and a smaller overshoot current. However, in the process of post-simulation, due to the influence of parasitic capacitors, the total capacitance in nodes A and B changes, which also changes the charging and discharging speed, and the mismatch time, unpredictably. Currently, we often adjust the switch transistor and current mirror transistor size relatively. However, the modification of the device parameters brings extra bias to the design. In the whole design process, modifying the schematic and layout repeatedly is usually necessary.

Using the prediction model proposed in this paper to predict the RC network at nodes A and B, and add it to the netlist of the pre-simulation, the influence of the back-end parasitic should be predicted in the pre-simulation stage; this would greatly reduce the design cycle.

### 3.2. Low-Voltage VCO

The voltage-controlled oscillator (VCO) [11–13] linearly converts the input control voltage into the output frequency. The circuit structure of the LC voltage-controlled oscillator in this work is shown in the Figure 4.

As the key nodes in the VCO, VCOP and VCON are responsible for differentially outputting the generated frequencies, they drive the next stages in the PLL, while being directly connected to the drain of the differential pair, variable capacitance, fixed capacitance,

and spiral inductance. As the VCO needs to have a certain tuning range, the two key nodes above often need to be directly connected with dozens of groups of capacitor arrays, also controlled by switches in practice.

The network is not only topologically complex, but also needs to span multiple different areas in the routing of the layout, resulting in complex parasitic capacitors and coupling capacitors; this makes a big difference between the pre-simulation and post-simulation without the parasitic parameters [14]. The output frequency of the VCO can be expressed as:

$$f_0 = \frac{1}{2\pi\sqrt{L\left(C_{var} + C_{fix} + C_{SCA} + C_{par}\right)}} \tag{1}$$

Here, $C_{var}$ is the variable capacitance, $C_{fix}$ is the fixed capacitance, $C_{SCA}$ is the capacitance of the capacitor array connected to the resonant network, and $C_{par}$ is the total parasitic capacitance of the output network, including the intrinsic parasitic capacitance and the coupling parasitic capacitance.

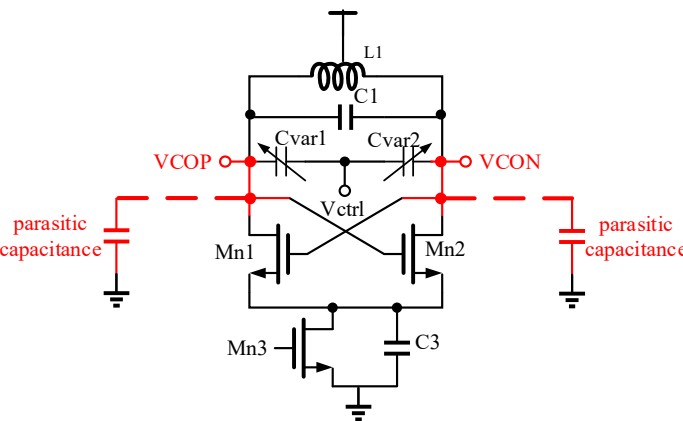

**Figure 4.** Class-C VCO.

As is demonstrated, when the two differential nodes above have a parasitic capacitance that is too large (most likely when the VCO tuning range is large with a large capacitance array), the equivalent fixed capacitance will exceed the proper value; this depresses the total capacitance range, causes a tuning gain drop, and reduces the overlap of adjacent frequency bands. As a result, some of the VCO frequency points are missing, the ability of continuous tuning is lost, and the function of PLL is affected. In some scenarios, the existence of parasitic capacitance will directly affect the size of the variable capacitance; this will further affect the tuning linearity and cause a change in the loop parameters of the PLL resulting in loop instability.

In the current design process, designers change device parameters according to simulation results, and gradually achieve a best balance between the capacitor value and transistor size. This process greatly relies on designers' experience; however, but the circuit parameter changes will lead to layout changes, resulting in extra change in the parasitic parameters. Therefore, parameter modification, based on parasitic parameter extraction and post-simulation, often requires several iterations [15].

By adopting the parasitic prediction model proposed in this paper, the parasitic parameter reference values of the VCOP and VCON can be given in the pre-simulation stage, and the designer can adjust the device parameters in the circuit diagram accordingly; this is in order to effectively reduce the gap between the pre-simulation and post-simulation, and approximate the ideal design results as far as possible, without many re-layout designs. In this work, the following process is designed to speed up the optimization of the VCO circuit parameters. The optimization method of the VCO is shown in Figure 5.

Among them, according to the simulation results for the starting vibration and the center frequency of the transient simulation, the fixed capacitance in the circuit and the parameters of the oscillating MOS are adjusted; the device parameters of the variable capacitor and switched capacitor array are adjusted according to the frequency coverage provided by Hb simulation. The voltage sensitivity is ensured by adjusting the variable capacitor value, and the frequency band spacing is achieved by adjusting the variable capacitor value to ensure that the lowest frequency of this frequency band can cover the midpoint of the adjacent frequency band, as shown in Figure 6.

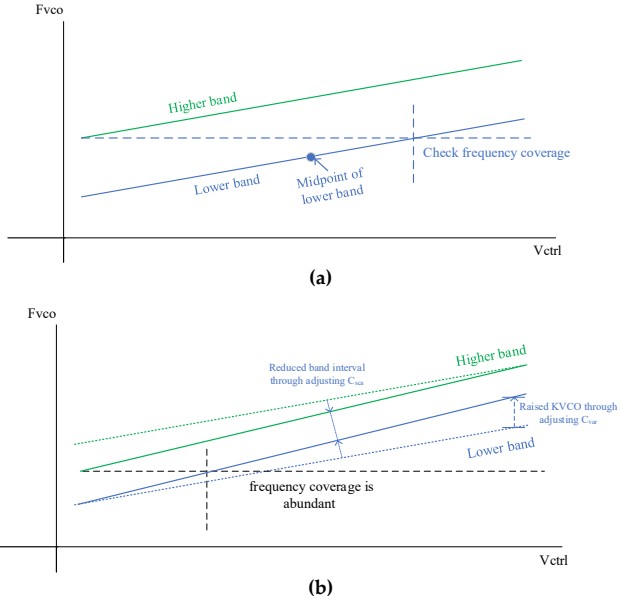

**Figure 5.** Optimization of a Class-C VCO (**a**) Frequency coverage is not enough as lowest frequency on next band should cover at least half the current bandwidth; (**b**) Optimizing the VCO design parameters to meet the design requirements.

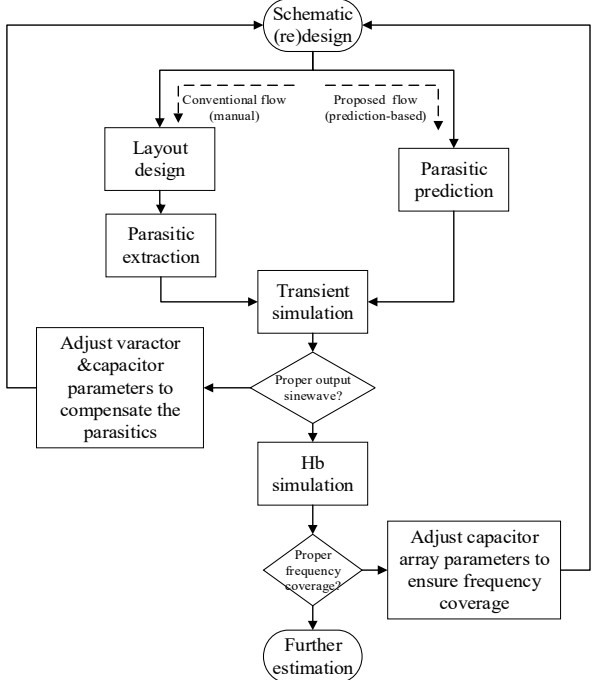

**Figure 6.** Optimization flow of the VCO.

## 4. Results and Analysis

Here we will present the specification of the circuit, optimized based on the proposed parasitic prediction, and compare it with the default process. We will focus on the price paid to achieve similar optimization results, as well as the circuits' performance after different optimization flows are completed.

### 4.1. Accuracy of the Model

Based on the prediction model proposed above, the key modules of the PLL underwent the TSMC 28 nm process; namely, the voltage control oscillator and charge pump underwent the parasitic prediction procedure. The prediction accuracy of the proposed method was compared with that of the post-simulation. In addition, the key performance of the optimized module was tested, including the tuning range of the VCO and the mismatch time of the charge pump.

The accuracy of the proposed prediction method was verified by comparing the predictions with the parasitic parameters obtained by RC extraction, and the results are displayed in Figure 7.

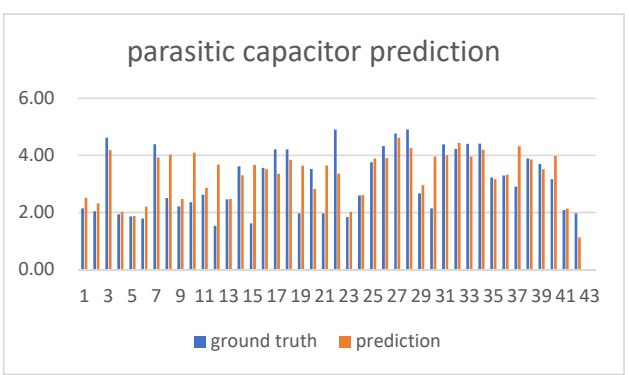

**Figure 7.** Capacitor prediction compared to tool-based RC extraction, the values are log-normalized to appear concise.

It can be observed from the figure that the prediction effect of the RC model is promising. The accuracy of the capacitance prediction is less than 20%, exponentially.

The prediction accuracy of the proposed model is compared with the data from the pre-simulation and MLParest [2]; the proposed method has significantly improved the accuracy compared to the pre-simulation, probably because the RC model used in this paper involved a depth graph that could better reflect the information of the whole circuit. Although comparing the accuracy of the model with [2] is not feasible due to the different technology adopted (10 nm&14 nm vs. 28 nm), the reproduced model indicated a minor accuracy improvement (20% vs 22% exponentially).

### 4.2. VCO Optimization Results

According to the prediction model proposed in this paper, the performance indicators of key modules in the PLL were optimized, and the outcome after each round of optimization is shown in Figure 8. As the proposed method does not involve layout design or verification, the design effort for each round of iteration is modeled as a quarter of that in a conventional manually designed iteration; each time, re-designing the layout is labor-concentrated.

As can be seen from the figure, after considering the design effort for each round of iteration, the performance of interest is improved with a much faster rate after joining the model-based optimization proposed in this paper; with this, the VCO frequency band overlap degree is improved by 6.8%, and the frequency band coverage ratio is improved by 2.3 percents.

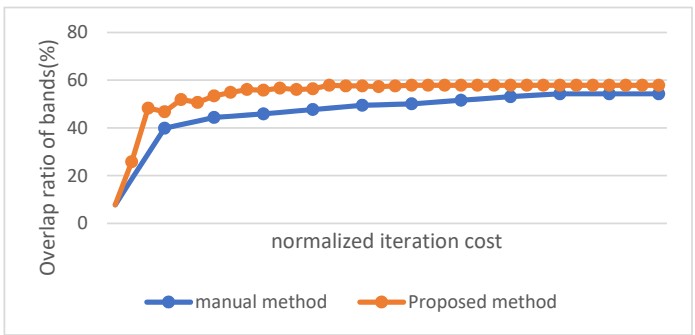

**Figure 8.** Optimization efficiency compared to a conventional flow on VCO.

### 4.3. CP Optimization Results

By improving the design flow of the CP circuit, the mismatch between the charge and the discharge loop can be contained in a smaller design and evaluation effort; in addition, the current overshoot and charge–discharge mismatch can be 12 ps better if the design time spent on it is equal, as shown in Figure 9. In this comparison, we chose to compare the design result after an equal effort of re-design and re-estimation. The proposed optimization flow, in this case, was able to perform more rounds of design parameter sets, leading to a broader design space exploration.

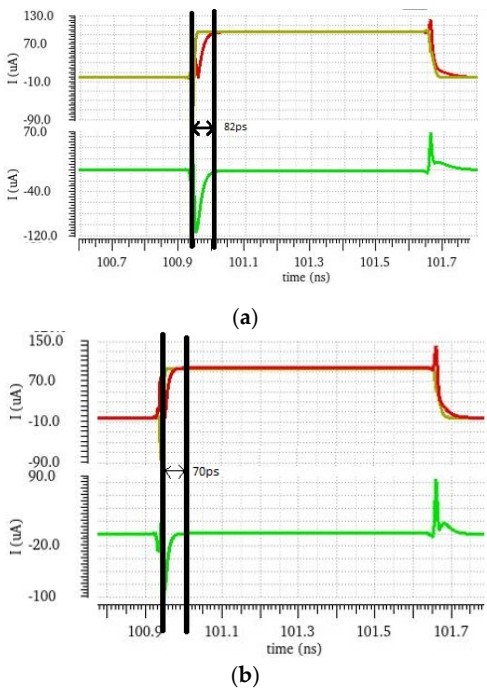

**Figure 9.** Comparison of optimized charge pumps: (**a**) after manually optimization; (**b**) after prediction-based optimization.

### 4.4. PLL Optimization Results

The new PLL circuit obtained by the proposed method has been improved in terms of loop characteristics and output noise in a more efficient fashion. These optimizations have been guided without a re-layout. As is demonstrated in Figure 10, they are a good addition to the existing design process. The following table shows the optimized PLL circuit parameters obtained by a final post-simulation. They are compared to the design results obtained under the traditional design process in Table 4. The layout of the final PLL circuit is shown in Figure 11.

**Table 4.** Optimization result.

| (sub)Circuit | Specification | After manual Optimization | After GNN-Aided Optimization |
|---|---|---|---|
| VCO | Adjacent band overlap rate | 17.1% | 19.4% |
| CP | Mismatch time | 82 ps | 70 ps |
| PLL | Phase noise@10 MHz | −96.17 dBc/Hz | −107.86 dBc/Hz |

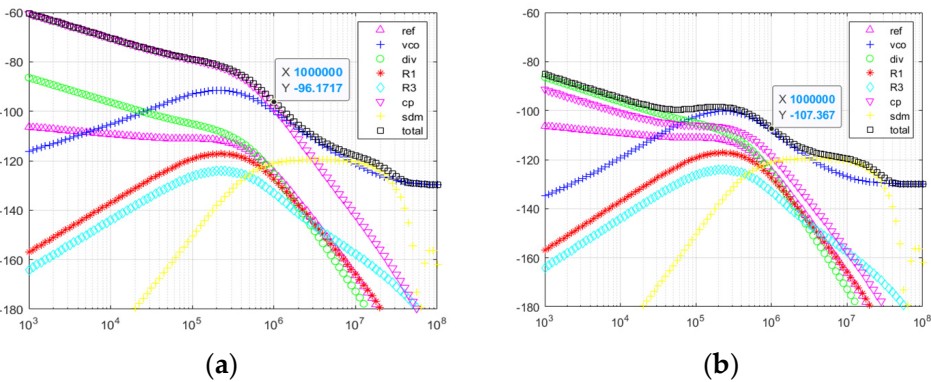

(**a**)　　　　　　　　　　　　　　　　(**b**)

**Figure 10.** Optimization result compared to a conventional flow on overall PLL: (**a**) manual optimization with limited iterations; (**b**) proposed optimization with more layout-free design iterations.

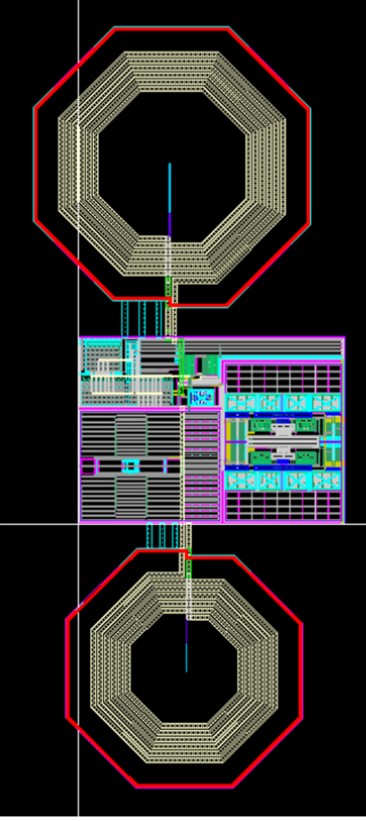

**Figure 11.** Layout of the final PLL circuit.

## 5. Conclusions

In this paper, a pre-layout parasitic parameter prediction model-based Graph Neural Network (GNN) is proposed for the optimal design of the PLL. State-of-art GNN models and methods are introduced to improvement the present model ([16–22]). The optimal designs are obtained by optimizing the specific design specifications of the CP and the

VCO in the PLL, such as the tuning range of the VCO and the mismatch current in the CP, etc. The proposed GNN prediction model is used to extract the predicted parasitic at key nodes in the schematic netlist so that the many variables needed to be fine-tuned are considered in the schematic design process; thus, the iteration times could be reduced significantly. In addition, the prediction model is applied to the design optimization of a 28 nm PLL for verification. This method has been proven to achieve the performance optimization of the PLL circuit module under the premise of greatly saving the optimization iteration and verification cost. The VCO designed by the above method only requires three iterations, which is similar to the frequency band superposition rate after 10 iterations in the conventional circuit design. Finally, the VCO band superposition rate increased by 2.3%. The CP optimal design is superior to the traditional method with a mismatch time of 12 ps.

**Author Contributions:** Methodology, C.L. and D.H.; Resources, X.Z.; Software, C.L.; Validation, C.L., D.H. and X.Z.; Writing—original draft, C.L.; Writing—review and editing, D.H. All authors have read and agreed to the published version of the manuscript.

**Funding:** This research was supported by the National Natural Science Foundation of China under Grant NSFC 61874152.

**Data Availability Statement:** The data presented in this study are available on request from the corresponding author. The data are not publicly available due to the circuit data are not published.

**Conflicts of Interest:** The authors declare no conflict of interest.

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
