# Peer review of "Pre-Layout Parasitic-Aware Design Optimizing for RF Circuits Using Graph Neural Network"

_electronics, doi:10.3390/electronics12020465_

Round 1

Reviewer 1 Report

1. Can the authors present a final schematic and layout of the designed PLL? Also, can the authors compare the resulting layout from the proposed optimization approach and that from the manual optimization approach?

2. The reviewer is concerned that the comparison between the proposed approach should be performed with other optimization approaches rather than the manual approach which seems to be ambiguous since the manual optimization strongly depends on the quality of the manual design. Have the authors compared the proposed method with other similar optimization methods in the reference?

Reviewer 2 Report

The paper presents a graph neural network (GNN) based approach towards parasitic estimation in analog / mixed signal circuit layouts. Two very relevant and interesting circuit examples, viz. an LC VCO and the charge pump components corresponding to the PLL are included and studied by the authors for parasitic estimation and subsequent response. The study is detailed and explains the approach well. Other than minor spell checks, for instance 'capasitor', instead of capacitor in Fig. 7 and the like, the reviewer does not have any major concerns.

Round 2

Reviewer 1 Report

My inquiries were clarified.